# Definition of a Threshold for the Plasma Aβ42/Aβ40 Ratio Measured by Single-Molecule Array to Predict the Amyloid Status of Individuals without Dementia

**DOI:** 10.3390/ijms25021173

**Published:** 2024-01-18

**Authors:** Lise Colmant, Emilien Boyer, Thomas Gerard, Kristel Sleegers, Renaud Lhommel, Adrian Ivanoiu, Philippe Lefèvre, Pascal Kienlen-Campard, Bernard Hanseeuw

**Affiliations:** 1Institute of Neuroscience, UCLouvain, 1200 Brussels, Belgium; lise.colmant@uclouvain.be (L.C.); emilien.boyer@uclouvain.be (E.B.); thomas.gerard@uclouvain.be (T.G.); philippe.lefevre@uclouvain.be (P.L.); pascal.kienlen-campard@uclouvain.be (P.K.-C.); 2Neurology Department, Cliniques Universitaires Saint-Luc, 1200 Brussels, Belgium; renaud.lhommel@uclouvain.be; 3Institute of Information and Communication Technologies, Electronics and Applied Mathematics, UCLouvain, 1348 Louvain-la-Neuve, Belgium; 4Complex Genetics of Alzheimer’s Disease Group, VIB-UAntwerp Center for Molecular Neurology, University of Antwerp, 2000 Antwerpen, Belgium; kristel.sleegers@uantwerpen.vib.be; 5WELBIO Department, WEL Research Institute, Avenue Pasteur, 6, 1300 Wavre, Belgium

**Keywords:** Alzheimer’s disease, plasma amyloid, SIMOA, plasma Aβ42/Aβ40 ratio, amyloid prediction, biomarkers, Alzheimer’s disease screening

## Abstract

Alzheimer’s disease (AD) is characterized by amyloid beta (Aβ) plaques and hyperphosphorylated tau in the brain. Aβ plaques precede cognitive impairments and can be detected through amyloid-positron emission tomography (PET) or in cerebrospinal fluid (CSF). Assessing the plasma Aβ42/Aβ40 ratio seems promising for non-invasive and cost-effective detection of brain Aβ accumulation. This approach involves some challenges, including the accuracy of blood-based biomarker measurements and the establishment of clear, standardized thresholds to categorize the risk of developing brain amyloid pathology. Plasma Aβ42/Aβ40 ratio was measured in 277 volunteers without dementia, 70 AD patients and 18 non-AD patients using single-molecule array. Patients (*n* = 88) and some volunteers (*n* = 66) were subject to evaluation of amyloid status by CSF Aβ quantification or PET analysis. Thresholds of plasma Aβ42/Aβ40 ratio were determined based on a Gaussian mixture model, a decision tree, and the Youden’s index. The 0.0472 threshold, the one with the highest sensitivity, was retained for general population without dementia screening, and the 0.0450 threshold was retained for research and clinical trials recruitment, aiming to minimize the need for CSF or PET analyses to identify amyloid-positive individuals. These findings offer a promising step towards a cost-effective method for identifying individuals at risk of developing AD.

## 1. Introduction

Alzheimer’s disease (AD) is the most common cause of dementia [1]. AD is characterized by the abnormal accumulation of amyloid beta (Aβ) peptides in the brain, and the aggregation of hyperphosphorylated tau deposits in neurons, forming neurofibrillary tangles. Aβ peptides are naturally released in the brain by the proteolytic cleavage of the amyloid-β precursor protein (APP) [2]. In AD, there is an imbalance between production and clearance of Aβ, which leads to Aβ accumulation in brain parenchyma. Longer isoforms of Aβ (e.g., Aβ42 or Aβ43) have intrinsic nucleation properties [3], leading to the formation of toxic oligomers that can further promote aggregation and fibrillization [4] to eventually form plaques in the brain [5]. The formation of toxic oligomers containing long forms of Aβ (Aβ42 or Aβ43) or fibrils typically containing Aβ42 results in decreased concentrations of soluble monomeric Aβ42 circulating in cerebrospinal fluid (CSF) [6]. The shorter and more abundant soluble Aβ40 form is less prone to aggregation, and its concentration in biofluids is assumed to distinguish high and low β-amyloid producers. The CSF soluble Aβ42/Aβ40 ratio decreases in AD, and the ratio may be more relevant to the diagnosis of AD than soluble Aβ42 or Aβ40 levels alone [7], because it normalizes the soluble Aβ42 concentration to intra-individual levels of Aβ production that are reflected by soluble Aβ40 concentration.

Positron emission tomography (PET) imaging studies showed that Aβ plaques precede the formation of tau neurofibrillary tangles and cognitive decline [8]. Individuals with abnormal amyloid deposits and no cognitive impairment can be characterized as having preclinical AD [9]. These clinically normal individuals with abnormal amyloid deposits have a higher risk of developing cognitive decline than individuals without amyloid deposition [10]. Determining amyloid status in clinically normal individuals thus allows identification of a population at risk of developing AD. Amyloid status (positive or negative) is currently determined by measuring Aβ42 in the CSF or by PET-based radiotracers detecting amyloid deposits in the brain parenchyma. However, their widespread use is restricted because they are invasive (CSF sampling) or expensive (PET).

The use of blood biomarkers, including Aβ measurement, has attracted growing interest [11]. They have long been inaccessible due to the low concentration of AD biomarkers (e.g., soluble Aβ42) in the blood and technical limitations due to the detection threshold of the tests classically used (enzyme-linked immunosorbent assay, ELISA) to monitor them in the CSF. Indeed, the concentrations of Aβ are 50–100 times lower in the plasma than in the CSF [12]. The recent emergence of novel technologies that allow protein detection in the femtomolar range (typically around or below 1 pg/mL) has sparked clinical interest in blood biomarkers. The low concentrations of plasma Aβ can now be quantified by approaches initially described as single-molecule enzyme-linked assay (or digital ELISA) and now often referred to as single-molecular array (SIMOA) [13,14]. Single-molecule analysis can be achieved using paramagnetic beads coupled with specific antibodies, enabling the binding of a single specific target (antigen) to one bead at low concentrations. These beads can then be loaded onto an array to measure the signal of a single antigen/antibody complex in the detection step. This approach has also been developed for multiplex (e.g., Aβ42/Aβ40) assays, allowing the measurement of different biomarkers in the same sample.

Because Aβ deposition in the brain is the earliest pathological signature of AD, measuring the soluble Aβ42/Aβ40 ratio appears to be promising for the detection of AD [15], even at preclinical stages. Blood measurement of Aβ has the advantage of being minimally invasive compared to lumbar puncture, and less expensive than PET imaging. Several studies have already shown that a decrease in the plasma soluble Aβ42/Aβ40 ratio was an indicator of brain amyloidosis [16,17,18,19] and could predict cognitive decline [20,21]. Existing tools include mass spectrometry and SIMOA assays (Quanterix^®^), with SIMOA being easier to implement in clinical research settings. 

Even though accurate measurement of the plasma soluble Aβ42/Aβ40 ratio has been reported as a potential tool for the detection of early AD stages, no cutoff value for this ratio has been clearly established in the literature to define amyloid status (positive or negative). The aim of the present study was to determine a threshold of the plasma soluble Aβ42/Aβ40 ratio measured by SIMOA in order to use this plasmatic measure as a screening tool for amyloid pathology, and to identify individuals at risk of developing AD before dementia. Individuals with plasma soluble Aβ42/Aβ40 ratios within the normal range would be characterized as free of amyloid pathology, and individuals with abnormal biomarker values would be recommended toward further confirmatory examination, including CSF biomarker assessments or PET imaging. This screening tool also holds promise for potential utilization in forthcoming treatments for AD, such as Lecanemab or Donanemab, both of which have demonstrated improved efficacy in mitigating cognitive decline in patients with prodromal AD or mild AD dementia [22,23]. The plasma soluble Aβ42/Aβ40 ratio could be used to measure the outcome of these treatments on amyloid pathology.

## 2. Results

### 2.1. Participants

We recruited 277 volunteers in Belgium, aged over 50 years and without known cognitive impairment, through local advertisements. All volunteers were without dementia. We also enrolled 88 patients from the Memory Clinic at Saint-Luc University Hospital, all of whom had reported memory complaints during a prior consultation. Patients had been clinically diagnosed with AD (*n* = 70) or non-AD (*n* = 18) neurodegenerative diseases. Demographic characteristics of the participants are presented in Table 1. 

AD patients were significantly older than volunteers (*p* < 0.0001) and non-AD patients (*p* = 0.017). Age was not different between volunteers and non-AD patients. As expected, Mini-Mental State Examination (MMSE) scores were higher among volunteers than among AD (*p* < 0.0001) and non-AD patients (*p* < 0.0001); MMSE scores were also higher among non-AD patients compared to AD patients (*p* = 0.0011). As expected, there were more apolipoprotein E (*APOE)* ε4 carriers among AD patients than volunteers (*p* < 0.0001), and among AD than non-AD patients (*p* = 0.0006). Indeed, the ε4 allele of the *APOE* gene increases the risk of developing AD [24,25,26]. The *APOE* ε4 status was not different between non-AD patients and volunteers. The gender repartition was not different between groups. Among AD patients, 23 individuals had dementia; among non-AD patients, 2 individuals had dementia.

### 2.2. Amyloid Status

#### 2.2.1. Amyloid Status of All Individuals

The cohort of 88 patients, along with a subset of volunteers (*n* = 73), underwent CSF analysis or an amyloid-PET scan for validation of their amyloid status. The final analysis included only individuals with a maximum time delay of 2 years between the blood test and CSF/PET amyloid status measurement unless the CSF/PET amyloid measurement was performed before the blood test and gave an amyloid-positive result. The final sample comprised 66 volunteers, 69 AD patients, and 16 non-AD patients.

Amyloid-PET or CSF analysis classified 74 participants as amyloid positive (Aβ+) and 77 participants as amyloid negative (Aβ−) (Table 2). Aβ+ individuals were significantly older than Aβ− individuals (*p* = 0.03). Gender repartition did not differ between these two groups. As expected, MMSE scores were significantly lower for Aβ+ participants than for Aβ− participants (*p* < 0.0001), and there were more ε4 carriers among Aβ+ participants than among Aβ− participants (*p* < 0.0001). Table 2 also shows the clinical diagnosis of participants (volunteers, AD, or non-AD patients), and the way that amyloid was measured (CSF, [18F]flutemetamol PET, or [11C]Pittsburg Compound B ([11C]PIB) PET).

#### 2.2.2. Amyloid Status of Individuals without Dementia

As we focused on the evaluation of the amyloid status of individuals without dementia, the sensitivity and specificity of tests were evaluated on this subgroup of individuals. The characteristics of Aβ+ and Aβ− individuals without dementia are described in Table 2. 

Aβ+ individuals without dementia were significantly older than Aβ− individuals without dementia (*p* = 0.017). Gender repartition did not differ between two groups. As expected, MMSE scores were significantly lower for Aβ+ participants than for Aβ− participants (*p* = 0.0002), and there were more ε4 carriers among Aβ+ participants than among Aβ− participants (*p* < 0.0001). Table 2 also shows the clinical diagnoses of participants (volunteers, AD, or non-AD patients), and the way that amyloid was measured (CSF, [18F]flutemetamol PET, or [11C]PiB PET).

#### 2.2.3. Distribution of the Plasma Aβ42/Aβ40 Ratio

The distribution of the plasma soluble Aβ42/Aβ40 ratio is represented for the different populations in Figure 1. When considering all individuals (volunteers, AD, and non-AD patients), the plasma Aβ42/Aβ40 ratio was lower for volunteers than AD patients (*p* < 0.0001), and lower for AD patients than for non-AD patients (*p* = 0.033) (Figure 1A). To determine the sensitivity and specificity of the plasmatic test for different thresholds, only individuals with a known CSF/PET amyloid status were considered (Figure 1B). The plasma Aβ42/Aβ40 ratio was lower for Aβ+ individuals than for Aβ− individuals (*p* < 0.0001). Finally, we restricted the analysis to individuals without dementia (Figure 1C). The plasma Aβ42/Aβ40 ratio stayed lower for Aβ+ individuals without dementia than for Aβ− individuals without dementia (*p* = 0.0002).

### 2.3. Determintation of the Threshold

The threshold for plasma soluble Aβ42/Aβ40 ratio that could be used for the classification of individuals as plasmatic Aβ+ or Aβ− was determined using three methods: an unsupervised learning method (Gaussian mixture model, GMM), a supervised learning method (decision tree), and the Youden’s index calculated on the Receiver Operating Characteristic (ROC) curve. The unsupervised method was applied to all participants, including those with no amyloid measure performed using CSF or PET. The supervised method included only participants with a known amyloid status using CSF or PET. The ROC curve was estimated for individuals with a known amyloid status and without dementia. 

#### 2.3.1. GMM-Based Classification of Plasma Aβ

All participants (volunteers, AD, and non-AD patients) were included in the GMM analysis except for eight individuals (2.2%) whose plasma soluble Aβ42/Aβ40 values were outliers. The GMM was modeled with two components, as our aim was to classify the participants into two groups (plasmatic Aβ+ or Aβ−) (Figure 2). The intersection of the two gaussian curves was localized at 0.0472; participants with a plasma soluble Aβ42/Aβ40 ratio ≤0.0472 were classified as plasmatic Aβ+, and those with a ratio >0.0472 were classified as plasmatic Aβ−.

Using this threshold, 95.7% of AD patients were accurately categorized as plasmatic Aβ+ (Table 3), suggesting the presence of amyloid pathology. This specific threshold aligns with the 95th percentile of AD patients. Conversely, 44.4% of non-AD patients were appropriately classified as plasmatic Aβ−, suggesting the absence of amyloid pathology. Finally, 58.5% of the volunteers were classified as plasmatic Aβ−, suggesting the absence of amyloid pathology. The true amyloid status of volunteers remains unknown, but it is possible that some volunteers would have amyloid brain deposits, corresponding to individuals with preclinical AD.

Among the volunteers who were ε4-carriers, 49.4% (41/83) were classified as plasmatic Aβ+; among the volunteers who were ε4-noncarriers, 38.5% (74/192) were classified as plasmatic Aβ+. The proportion of plasmatic Aβ+ and plasmatic Aβ− volunteers did not differ significantly between ε4-carriers and noncarriers (*p* = 0.12).

The sensitivity and specificity of the plasma soluble Aβ42/Aβ40 ratio were evaluated using a subgroup of participants with known amyloid status. Table 4 shows the number of participants classified as plasmatic Aβ+ or plasmatic Aβ− according to their CSF or PET amyloid status, using the 0.0472 threshold determined by GMM. Sensitivity was 87.8% and specificity was 55.8%.

Since our focus lies in ascertaining the amyloid status of individuals without documented cognitive impairment, we conducted an evaluation of the test’s sensitivity and specificity for classifying participants without dementia (Table 5), using the same threshold value of 0.0472. The sensitivity was 83%, and the specificity was 57.5%.

#### 2.3.2. Decision Tree

A decision tree was used as a second independent method to determine a suitable threshold for the plasma soluble Aβ42/Aβ40 ratio. A decision tree with one split was trained on one feature: the plasma soluble Aβ42/Aβ40 ratio. The threshold determined by the decision tree was 0.0465 (Figure 3), and it yielded a sensitivity of 86.5% and specificity of 58.4% (Table 6).

Subsequently, individuals without dementia were analyzed to determine whether those classified as plasmatic Aβ+ were indeed Aβ+ and whether those classified plasmatic Aβ− were indeed Aβ− (Table 7). For participants without dementia, the threshold value of 0.0465 yielded a sensitivity of 81.1% and a specificity of 60.3%.

#### 2.3.3. ROC Curve

We plotted a ROC curve using only individuals without dementia for whom the amyloid status was defined by CSF or PET examination (Figure 4). The area under the curve (AUC) was 0.728 (95% confidence interval (CI): 0.642–0.815). Youden’s index was calculated to maximize the sum of sensitivity and specificity and it corresponded to a cutoff value of 0.0450. Using the Youden’s index, sensitivity was 75.4% and specificity 67.1% for individuals without dementia.

### 2.4. Posttest Probabilities

Two cutoff values were retained: the one giving the best sensitivity (0.0472) and the one giving the best specificity (0.0450). The posttest probabilities of having brain amyloidosis were calculated according to the plasma soluble Aβ42/Aβ40 ratio at these two cutoff values, and for two different ages (60 and 80). The prevalence of amyloid pathology among the clinically normal population was 15.8% at 60 years old [27], corresponding to pretest odds of 0.187. The prevalence of amyloid pathology among the clinically normal population at age 80 was 32.6% [27], corresponding to pretest odds of 0.484.

The threshold determined by GMM (0.0472) gave the best sensitivity. At this threshold, and among the population without dementia, the sensitivity was 83.0%, specificity was 57.5%, LH+ was 1.95, and LH− was 0.29. At age 60, if the plasmatic test gave a positive result then the posttest odds were 0.365, corresponding to a posttest probability of 26.8%; if the plasmatic test gave a negative result then the posttest odds were 0.055, corresponding to a posttest probability of 5.2%. At this same threshold at age 80, a positive plasmatic test gave a posttest probability of brain amyloidosis of 48.6%, and a negative plasmatic test gave a posttest probability of 12.5%.

The Youden’s index (0.0450) gave a specificity of 67.1%, a sensitivity of 75.4%, LH+ of 2.30, and LH− of 0.36. With this threshold, at age 60, if the plasmatic test gave a positive result then the posttest probability of having brain amyloidosis was 30.0%; if the plasmatic test gave a negative result then the posttest probability was 6.4%. At this same threshold, at 80, a positive plasmatic test gave a probability of brain amyloidosis of 52.6%, and a negative plasmatic test of 15.0%.

## 3. Discussion

The present study confirmed that plasmatic measures of Aβ are useful for predicting cerebral Aβ deposition [14,17,19,28,29], as measured using either in CSF or by PET analysis. Several blood Aβ assay methods have been developed recently, including mass spectrometry. The performances of plasma Aβ42/40 assays in detecting cerebral amyloid pathology in early AD patients was recently compared [30]. SIMOA and immunoprecipitation coupled to mass spectrometry (IPMS) appear to be the two assays whose sensitivity (detection level) is sufficiently good and that can be operated for large-scale sample assays [17,18,31]. We used SIMOA 3-Plex A assays, a promising technique that is much easier to implement routinely and with a higher throughput for measurement over a wider sample range than mass spectrometry, although its performance has been reported to be more modest than IPMS for the discrimination of AD vs MCI groups [32].

Biomarker concentration values alone are of limited use for classifying individuals in a cohort and predicting their clinical status, as no clear threshold was available in the literature. The aim of this work was to determine a threshold to use for classifying participants as amyloid positive or negative based on the plasma soluble Aβ42/Aβ40 ratio obtained by SIMOA [16]. We determined two different thresholds for use according to the context in which the plasmatic test would be used. The first threshold, which minimizes false negatives, is intended to be used as an AD exclusion tool; the second, which maximizes accuracy, is intended to be used to improve enrollment in AD research studies. 

In this study, a threshold of 0.0472 was used for the plasma soluble Aβ42/Aβ40 ratio measured by SIMOA, with the specific aim of applying it as a screening test to exclude an AD diagnosis in clinical practice. This threshold yielded the best sensitivity (83.0%), and a specificity of 57.5% when tested on adults without dementia. This threshold corresponds to the 95th percentile among AD patients. In clinical conditions, the selection of a cutoff value should prioritize sensitivity because missing any diagnosis is undesirable. The blood test would be the first examination performed, and more specific examinations would be ordered only if necessary. A sensitivity of 83.0% would reassure individuals with a negative plasma test that they are not at risk of developing AD. This threshold reduces the probability of having brain amyloid deposit for individuals receiving a negative test result to 5% at age 60, and to 12% at age 80. Therefore, in the case of a negative blood test result, individuals can be reassured in their sixties, and somewhat reassured in the eighties. Conversely, the low specificity means that specific assessments, such as amyloid-PET scans or CSF analysis, will be necessary follow-up for positive cases.

The threshold of 0.0450, obtained using the Youden’s index, improved specificity to 67.1% but reduced sensitivity to 75.4%. this method gave the best accuracy. A threshold with higher specificity could be useful for research applications, especially for enrollment in AD studies. It could be used to classify a large cohort of individuals without requiring PET imaging or CSF analysis. It could also be a helpful inclusion criterion in clinical trials studying amyloid in early stages of AD. Such prescreening for clinical trials recruitment would reduce the cost of clinical studies. To include one amyloid-positive individual aged 60 years old, an average of 6.3 amyloid-PET scans or CSF analyses are required given the known prevalence of amyloid positivity at this age in the general population (15.8%) [27]. In practice, using PET scans for screening is not recommended. With the availability of a prescreening blood test, the frequency of amyloid-PET scans or CSF analyses could be substantially reduced by limiting the use of these examinations to at-risk individuals. Adults aged 60 years old with a positive blood test defined by our criteria have a 26.8% probability of cerebral amyloid deposit. Therefore, if we aim to recruit one amyloid-positive individual aged 60 years old in a clinical trial, we need to perform 3.73 PET amyloid-PET scans (or CSF analyses) among individuals with a positive blood test, which is half the number required without a prior blood test (6.3 CSF/PET analysis).

The present findings expanded previous results on plasma amyloid measurements [14,17,19,29], and defined thresholds for the plasma soluble Aβ42/Aβ40 ratio measured by SIMOA. Other authors have reported correlations between PET or CSF amyloid status and the plasma soluble Aβ42/Aβ40 ratio measured by SIMOA [14,16,33]. Verberk et al. showed that the plasma soluble Aβ42/Aβ40 ratio could serve as a prescreening test for AD pathology, and that a lower plasma soluble Aβ42/Aβ40 ratio was associated with an increased risk of clinical progression to mild cognitive impairment or dementia [14]. These studies found that the best cutoff values to predict amyloid status varied from 0.038 [33] to 0.056 [34], without consensus. They focused on clinically unimpaired individuals as in our study. However, participants’ mean ages were higher than in the present cohort in two of the studies (mean age: 77) [33,34] and younger than our cohort in one study (mean age: 61) [14]. Additional studies investigating the effect of age on plasma soluble Aβ42/Aβ40 ratio should be carried out in the future. Our results confirmed a cutoff value between 0.038 and 0.056 in an independent cohort and extended it to a general population without dementia aged over 50. We retained a threshold of 0.0472 for the exclusion of AD, and a threshold of 0.0450 for research applications. 

A previous study evaluated plasma biomarkers measured by SIMOA against brain autopsy results [35]. The plasma soluble Aβ42/Aβ40 ratio decreased with brain amyloid deposits, but with only modest accuracy (AUC = 0.60). Immunoprecipitation followed by spectrometry analysis also allowed the measurement of plasma soluble Aβ42/Aβ40 ratio, and gave better sensitivity and specificity (AUC = 0.75–0.89) than immunoassays for plasma [15,17,28,33,36,37,38,39]. Although IPMS could appear as a future gold standard, some hurdles remain for its implementation as a routine technique in comparison to SIMOA. Other commercially available ELISA kits (EUROIMMUN, Lübeck, Germany) are capable of measuring soluble amyloid in plasma and demonstrate similar correlations to CSF when compared to SIMOA for Aβ42/40 ratio (AUC = 0.69–0.70) [30]. ELISA also exhibits comparable accuracy to SIMOA in detecting cerebral amyloidosis based on plasma amyloid levels (ELISA: AUC= 0.78, 95% CI 0.72–0.84; SIMOA: AUC = 0.79, 95% CI 0.73–0.85) [40]. Although the plasma Aβ42/40 threshold defined in the present study could be applied to assays performing similarly to SIMOA for quantifying plasma Aβ, it should be reconsidered for use with techniques that offer more precise measurement of different Aβ forms in plasma, such as IPMS [30].

The present study evaluated the plasma soluble Aβ42/Aβ40 ratio, but not other AD blood biomarkers such as total-tau or neurofilament light chains, as our aim was to have a measure of brain amyloidosis. Blood-based biomarkers of phosphorylated tau (p-tau231, p-tau181, and p-tau217) have recently become available for SIMOA. P-tau181 and p-tau231 showed good accuracy for predicting AD-related neuropathological changes, but this may reflect different brain processes, such as soluble tau pathology, and/or neural damage [35]. Plasma p-tau231 has been shown to increase in the early stages of AD but recent studies showed that p-tau217 better captures the earliest cerebral changes related to AD [41]. Prior studies have also shown a relationship between plasma p-tau181 and amyloid PET, even in clinically unimpaired participants [35,42,43]. Plasma p-tau181 could be an accurate predictor of AD [42]. However, only the soluble Aβ42/Aβ40 ratio relates purely to brain amyloidosis without considering tau pathology. Given our objective to predict brain amyloidosis rather than brain tauopathy in AD, our focus was solely on quantifying plasma amyloid levels. The associations between Aβ and p-tau measured in plasma require further evaluation in future studies.

Not all plasma amyloid peptides come from the central nervous system, which could explain the limited ability of observing AD-related Aβ pathology in plasma. The key components of Aβ-processing pathways are expressed in diverse extracerebral tissues, including the pancreas, appendix, gastrointestinal tract, and both male and female reproductive organs [44]. This suggests a potential peripheral Aβ production. Furthermore, compelling evidence links Aβ with the vascular system: higher plasma levels of Aβ have been associated with cerebral white matter lesions, cerebral microbleeds, hypertension, diabetes, and ischemic heart disease [16]. Aβ is also synthesized by the skeletal muscles, platelets, and vascular walls [45]. In addition, plasma proteins such as immunoglobulins, albumin, and complement are known to bind to and mask Aβ peptides [46]. Plasma Aβ is excreted by the kidneys, meaning that its level will rise in cases of renal dysfunction [47]. All of these parameters influencing amyloid levels in plasma could explain the challenges in achieving high specificity and sensitivity with blood-based amyloid tests.

One limitation of the present study is that the thresholds defined were established in participants older than 50. Because amyloid pathology is rare before age 50 [48], we do not recommend testing individuals for plasmatic amyloid before this age. Furthermore, aging has been reported to increase plasma Aβ in both humans [47] and nonhuman primates [49]. Additional studies are needed to determine a threshold for the plasma amyloid level for a population younger than 50 years old, and to evaluate whether the threshold should be adapted according to age.

Another limitation is that amyloid status was confirmed using two different methods: PET and CSF analysis. Although it would have been more consistent to use a single reference technique, it would have restricted the number of participants eligible for the study. Finally, soluble Aβ40 concentration was not measured in the CSF of our participants, in accordance with hospital protocols at the recruitment site. Given that the CSF soluble Aβ42/Aβ40 ratio enhances the accuracy of AD diagnosis by adjusting Aβ42 values based on whether the patient is a high or low amyloid producer (with a high or low quantity of Aβ40) [7], the absence of this measurement in our study may contribute to the fact that some patients received a clinical diagnosis of AD while being classified as Aβ− in CSF assessments. 

## 4. Materials and Methods

### 4.1. Participants

We recruited 277 volunteers in Belgium, aged over 50 years, and without known cognitive impairment. In order to validate the absence of cognitive impairment, volunteers completed the MMSE [50]. All volunteers had an MMSE score ≥24/30 and were thus without dementia. All volunteers reported no recent illness or change in medical treatment during the last three months. The exclusion criteria were alcohol abuse, active cancer, a major neurological condition like stroke, or a known neurodegenerative disorder. We also enrolled 88 patients from the Memory Clinic at Saint-Luc University Hospital, all of whom had reported memory complaints during a prior consultation and received a clinical diagnosis of AD (*n* = 70), or non-AD neurodegenerative disease (*n* = 18). The non-AD pathologies included: frontotemporal lobar degeneration, normal pressure hydrocephalus, microvascular disease, corticobasal degeneration, Lewy body dementia, or primary progressive aphasia.

This study was conducted in accordance with the Declaration of Helsinki and was approved by the institution’s Ethical Committee. All participants provided written informed consent (UCL-2022-473; UCL-2016-121; UCL-2018-119).

All participants (volunteers and patients) undertook a blood test after which deoxyribonucleic acid (DNA) and plasma were extracted from the blood sample. The plasma Aβ42/Aβ40 ratio was measured using SIMOA (Neurology 3 plex A, Quanterix). The genetic risk for AD was estimated by genotyping the apolipoprotein E gene (*APOE*).

The cohort of 88 patients, along with a subset of volunteers (*n* = 73), also underwent CSF analysis or an amyloid-PET scan for validation of their amyloid status. The final analysis included only individuals with a maximum time delay of 2 years between the blood test and CSF/PET amyloid status measurement. The blood test could have been performed before or after the CSF/PET amyloid status measurement. Participants were excluded if more than 2 years had elapsed between the two exams, unless the CSF/PET amyloid measurement was performed before the blood test and gave an amyloid-positive result (*n* = 21). Since none of the individuals in our cohort received anti-amyloid therapy, those who were previously identified as amyloid-positive would continue to retain their amyloid-positive status [51]. Six individuals were excluded who had a negative CSF/PET amyloid measurement performed more than 2 years before the blood test, and four were excluded who had a blood test performed more than 2 years before the CSF/PET amyloid status measurement. The final sample comprised 66 volunteers, 69 AD patients, and 16 non-AD patients.

Individuals were classified as amyloid positive (Aβ+) when the CSF amyloid measurement was lower than 438 pg/mL [52] or when the amyloid-PET Centiloid score was higher than 26 [53]. Otherwise, they were categorized as amyloid negative (Aβ−). Both volunteers and patients underwent an MMSE evaluation on the same day as the blood test. They were classified as having dementia when their MMSE score was below 24/30, and without dementia when their MMSE score was higher than or equal to 24/30.

### 4.2. Blood Drawing and Plasma Prepatation

A standard venipuncture procedure was performed using a 21 g needle, and blood was collected in ethylenediaminetetraacetic acid (EDTA) polypropylene K2 tubes (K2E K2EDTA, Vacuette Tube, REF: 455045). The tube was placed on ice immediately after collection and plasma isolation was performed within 2h. Blood was centrifuged at 2000× *g* for 10 min at 4 °C, and plasma was aliquoted at a volume of 500 µL into cryotubes then stored at −80 °C until further analysis.

### 4.3. APOE Genotyping

DNA was extracted from blood samples. Participants were analyzed for *APOE* polymorphisms rs429358 (which is a [T/C] substitution on chr19:44908684 (GRCh38.p14) of the sequence GCTGGGCGCGGACATGGAGGACGTG[T/C]GCGGCCGCCTGGTGCAGTACCGCGG), and rs7412 (which is a [C/T] substitution on chr19:44908822 of the sequence CCGCGATGCCGATGACCTGCAGAAG[C/T]GCCTGGCAGTGTACAGGCCGGGGC). Based on two single-nucleotide polymorphisms, alleles were assigned as ε2, ε3, or ε4. The *APOE* ε4 allele represents a major risk factor for AD [24,25,26]. We classified participants into two groups: “ε4 carriers” (ε3ε4, ε4ε4, and ε2ε4) and “ε4 noncarriers” (ε2ε2, ε2ε3 and ε3ε3); ε4 carriers have a higher risk of developing AD compared to ε4 noncarriers.

### 4.4. Quantification of Plasma Aβ40 and Aβ42

Quantification of soluble Aβ40 and Aβ42 was performed using the Neurology Plex A kit from Quanterix (Neurology 3 Plex A, REF:101995). Each plasma sample was thawed to room temperature for 1h before being processed for use in SIMOA, an assay that relies on distinct antibodies for capturing and detecting amyloid-β species (Aβ42, Aβ40). The capture antibody (6E10) recognizes the N-terminal region of both species (amino acids 4 to 10), whereas the detection antibodies are specific to the C-terminal ends of Aβ42 and Aβ40 to reveal them. Aβ42 and Aβ40 were simultaneously measured following the manufacturer’s instructions. The same experimenter and same duration of experiment were used for all assay runs. 

### 4.5. CSF Analysis

Aβ42 measurements in CSF were performed using Lumipulse^®^ (G β-Amyloid 1–42, REF: 230336) in the Biology Lab of the Saint-Luc University Hospital, Brussels, Belgium. Participants were classified as Aβ+ when CSF Aβ42 was below 438 pg/mL, and as Aβ− otherwise [52].

### 4.6. Amyloid-PET–CT Acquisition and Computing

Two amyloid-PET radiotracers were used in this study: [18F]flutemetamol and [11C]Pittsburg Compound B ([11C]PiB). An anatomical 3D-T1 MRI was acquired for each participant using a 3T MRI (GE Signa Premier, GE Healthcare, Chicago, IL, USA).

For both radiotracers, semi-quantitative PET data were computed using PNEURO software (v4.1) (PMOD LLC Technologies, Zurich, Switzerland) following the previously developed Centiloid pipeline [53] to return a Centiloid value for each participant. A Centiloid threshold of 26 was used to discriminate between participants considered Aβ+ (Centiloid > 26) and Aβ− (Centiloid ≤ 26). This threshold was previously shown to be the most useful for distinguishing patients who would progress to dementia from those who would remain clinically stable six years after PET imaging [53].

#### 4.6.1. [18F]flutemetamol PET-CT

Ninety minutes after [18F]flutemetamol (GE Healthcare, Chicago, IL, USA) intravenous injection (target activity 185 ± 5 MBq), a 30-min list-mode PET/CT acquisition was performed using a Philips Gemini TF (Philips Healthcare, Amsterdam, The Netherlands). The images were reconstructed as a dynamic scan of 6 × 5 min frames with 2 mm isometric voxels, including attenuation, scatter, and decay corrections, in addition to time-of-flight information using the manufacturer’s standard reconstruction algorithm. No partial volume correction was applied to the data.

#### 4.6.2. [11C]PiB PET-CT

Forty minutes after [11C]PiB intravenous injection (target activity 500 MBq), a 20 min list-mode PET/CT acquisition was performed using a Philips Vereos digital PET (Philips Healthcare, Amsterdam, The Netherlands). Images were reconstructed in 4 × 5 min frames with 2 mm isometric voxels using the manufacturer’s reconstruction algorithm that includes attenuation, scatter, and decay corrections, in addition to time-of-flight information.

### 4.7. Basic Characteristics

#### 4.7.1. Characteristics of Volunteers and Patients

Baseline characteristics were compared between volunteers and patients using Chi-square tests for gender and *APOE* ε4 status, and using ANOVA for age and MMSE, with posthoc *t*-tests performed and Holm’s correction for multiple testing applied where appropriate. 

#### 4.7.2. Characteristics of Participants with Amyloid Measured Using CSF or PET Analysis

A subgroup of volunteers and patients benefited from an amyloid measurement from either an amyloid-PET or CSF analysis, and they were classified as Aβ+ or Aβ−. The subgroup of volunteers to whom we offered PET or CSF analysis was enriched in *APOE* ε4 carriers, matching the frequency of ε4 carriage observed in AD patients. We compared characteristics of Aβ+ and Aβ− individuals, using t-tests for age and MMSE, and using Chi-squared tests for *APOE* ε4 status and gender repartition. We also compared characteristics of Aβ+ and Aβ− individuals without dementia using the same tests. 

#### 4.7.3. Distribution of the Plasma Aβ42/Aβ40 Ratio

We represented the distribution of the plasma Aβ42/Aβ40 ratio for the different populations (all individuals, all individuals with a CSF/PET known amyloid status, and all individuals without dementia with a CSF/PET known amyloid status). We visually verified using qq-plots that all distributions were normally distributed. The plasma Aβ42/Aβ40 ratio was compared between volunteers, AD patients, and non-AD patients using ANOVA; an equivalent comparison was made between Aβ+ and Aβ− individuals with *t*-tests.

### 4.8. Determination of Thresholds

A threshold for plasma Aβ42/Aβ40 ratio to be used to classify individuals as plasmatic Aβ+ or Aβ− was determined using three methods: an unsupervised learning method (GMM), a supervised learning method (decision tree), and the Youden’s index calculated on the ROC curve. 

#### 4.8.1. Gaussian Mixture Model (GMM)

A GMM was applied on the distribution of the plasma Aβ42/Aβ40 ratio of all participants (277 volunteers, 70 AD patients, and 18 non-AD patients) to approximate the distribution as two gaussian curves. The GMM analysis was implemented in R (version 4.2.1) using the mclust package (version 6.0.0) [54]. First, outliers were removed from the plasma Aβ42/Aβ40 distribution of all participants (*n* = 8, 2.2%). Outliers were defined according to an interquartile range (IQR) criterion; values were excluded if they fell above q_0.75_ + 1.5 IQR or below q_0.25_ − 1.5 IQR, where q_0.75_ and q_0.25_ corresponded to first and third quartile, respectively, and IQR was the difference between the third and first quartile.

The plasma Aβ42/Aβ40 distribution, without outliers, was then modeled using a two-component GMM with unequal variance. The threshold value of the Aβ42/Aβ40 ratio, used to categorize participants, corresponds to the point of intersection between the two Gaussian distributions. Participants with a plasma Aβ42/Aβ40 ratio below the threshold were considered plasmatic Aβ+ and those with a ratio above the threshold were considered plasmatic Aβ−.

The plasmatic amyloid classification of participants was then compared across clinical diagnoses (AD, non-AD, and volunteers). Among the volunteers, plasmatic amyloid classification was also compared between ε4-carriers and noncarriers using a Chi-squared test.

We also evaluated the sensitivity and specificity of the plasma test using the 151 participants (66 volunteers, 69 AD patients, and 16 non-AD patients) for whom amyloid status was determined by PET or CSF analysis. Finally, as we were specifically interested in the ability of the plasmatic measure to determine the amyloid status of individuals before dementia, we evaluated the sensitivity and specificity of the test based only on participants without dementia (*n* = 126) but with a known amyloid status.

#### 4.8.2. Decision Tree

The 151 participants with a known amyloid status were classified by a supervised algorithm. A decision tree was trained in MATLAB using the Classification Learner app, with a maximum of one split to classify individuals as plasmatic Aβ+ or plasmatic Aβ− based on one feature: the plasma Aβ42/Aβ40 ratio. We trained the model using all data. The decision-tree-split nodes were based on the Gini diversity index. This decision tree provided a classification threshold. The sensitivity and specificity of this method were estimated.

Finally, the sensitivity and specificity of the test were evaluated based only on individuals without dementia but with a known amyloid status.

#### 4.8.3. ROC Curve

We generated a ROC curve for the 126 non-demented individuals with confirmed amyloid status using R (version 4.2.1). In this analysis, we exclusively considered individuals without dementia to assess the plasma Aβ42/Aβ40 ratio’s capability to accurately identify amyloid status when used as a screening tool among individuals prior to dementia development. We computed the AUC and its 95% confidence interval. We determined the Youden’s index. We evaluated the sensitivity and specificity of test at this cutoff value.

### 4.9. Posttest Probabilities

Finally, posttest probabilities resulting in a positive or negative result of the plasma measurement were computed using R (version 4.2.1), with the likelihood ratio for the cutoffs giving the best sensitivity or specificity for clinically normal adults aged 60 or 80. The prevalences of amyloid pathology in the clinically normal population at these two ages (15.8% at 60, 32.6% at 80) were used to estimate the pretest probabilities [27].

The likelihood ratio for a positive test result (LR+) was defined as the sensitivity divided by one minus the specificity (LH+ = Sn/(1 − p)), and the likelihood ratio for a negative test result (LR−) as one minus the sensitivity divided by the specificity (LH− = (1 − Sn)/Sp). Then, the pretest odds were calculated as the pretest probability divided by one minus the pretest probability. Posttest odds were obtained by multiplying the pretest odds by the likelihood ratio. Finally, posttest odds were converted into the posttest probability.

## 5. Conclusions

The present findings build upon and refine previous work on plasma amyloid measurements. More specifically, this study established two plasma Aβ42/Aβ40 ratio thresholds measured by SIMOA, that were evaluated on a population without dementia aged over 50. The first threshold was 0.0472, offering higher sensitivity to minimize the risk of missing AD cases. In cases with positive results, individuals would benefit from additional exams such as PET or CSF analysis to confirm their amyloid status. The second threshold, 0.0450, may be more suitable for research and clinical trial recruitment to streamline processes and reduce costs. The significance of this threshold lies in its potential to facilitate early detection of amyloid pathology. These findings offer a promising step towards more accessible and cost-effective methods for identifying individuals at risk of developing AD, with potential implications for early intervention and clinical trial recruitment.

## Figures and Tables

**Figure 1 ijms-25-01173-f001:**
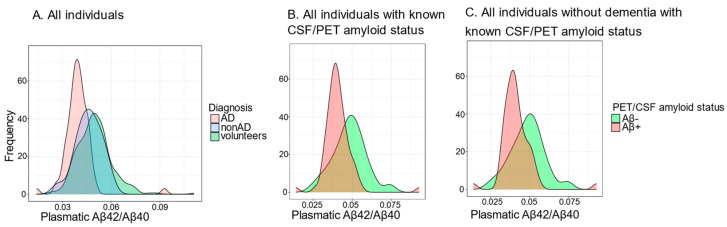
Distribution of the plasma soluble Aβ42/Aβ40 ratio among (**A**) all participants based on their clinical status; (**B**) among all participants with known amyloid status; and (**C**) among participants without dementia and with known amyloid status.

**Figure 2 ijms-25-01173-f002:**
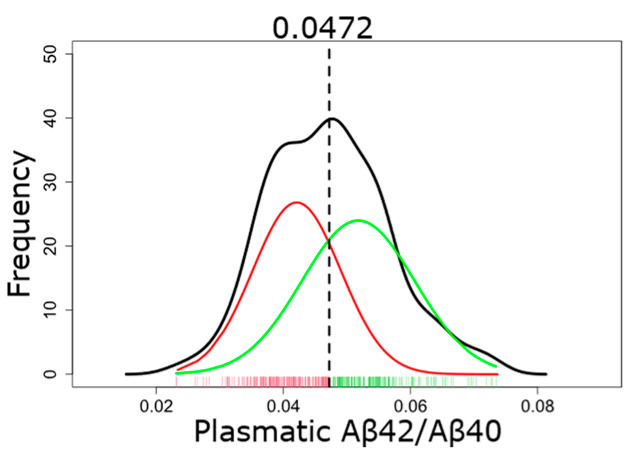
The full distribution of the plasma soluble Aβ42/Aβ40 ratio is shown in black. This distribution was then approximated as two gaussian curves, represented in red (plasmatic Aβ+) and green (plasmatic Aβ−). Each tick on the x-axis represents one individual. The threshold at the intersection of the two gaussian curves is 0.0472.

**Figure 3 ijms-25-01173-f003:**
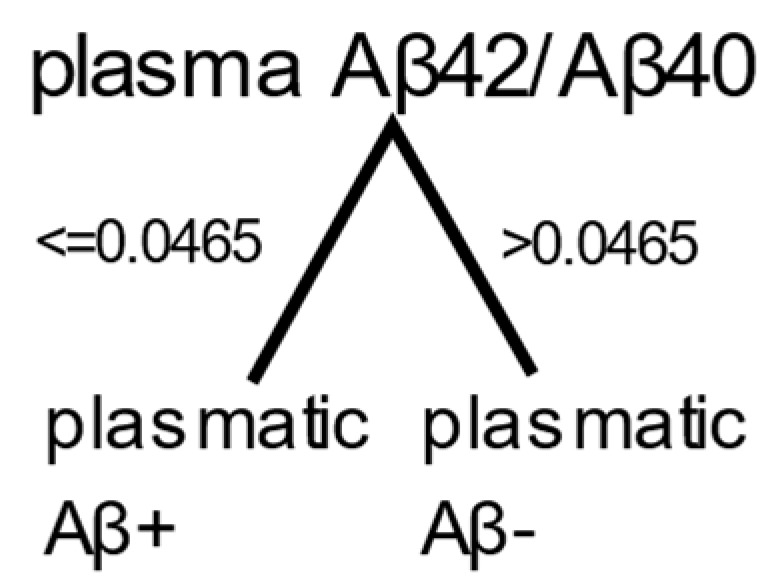
Result for the decision tree. Participants were classified as plasmatic Aβ+ if their plasma soluble Aβ42/Aβ40 ratio was ≤0.0465, and as plasmatic Aβ− otherwise.

**Figure 4 ijms-25-01173-f004:**
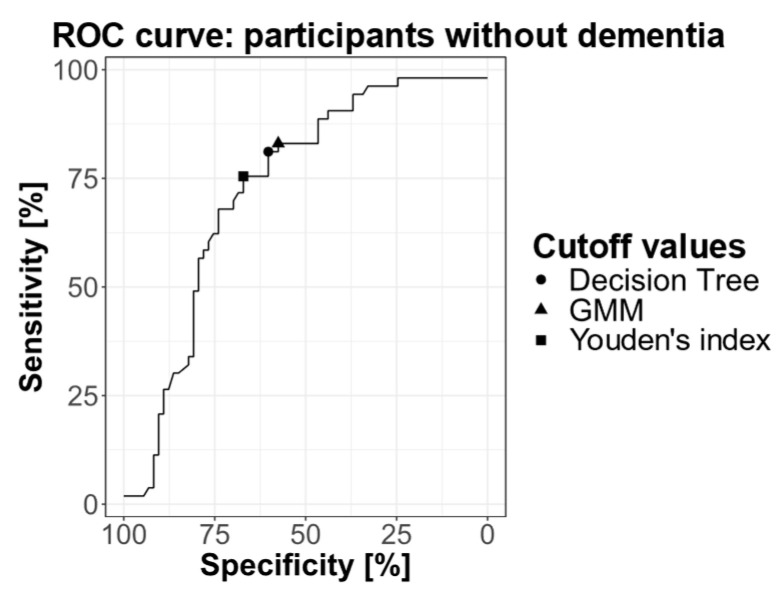
ROC curve representing the sensitivity and specificity of the plasma soluble Aβ42/Aβ40 ratio for different cutoff values, calculated on participants without dementia. The cutoff values obtained by GMM (0.0472) and by decision tree (0.0465) are indicated on the curve, and the Youden’s index is also shown (0.0450).

**Table 1 ijms-25-01173-t001:** Demographic characteristics of volunteers and patients. SD: standard deviation.

	Volunteers	AD Patients	Non-AD Patients
n	277	70	18
Age: mean (SD)	66.5 (7.78)	71.1 (8.05)	65.6 (9.55)
MMSE: mean (SD)	28.5 (1.27)	24.1 (4.59)	26.4 (2.12)
*APOE* *: n ε4−/n ε4+ (% ε4−/% ε4+)	192/83 (70%/30%)	21/40 (34%/66%)	14/2 (87.5%/12.5%)
Gender: male/female	98/179	32/38	7/11

* *APOE*: 2 missing among volunteers, nine missing among AD patients, 2 missing among non-AD patients; ε4− represents ε4 noncarriers and ε4+ represents ε4 carriers.

**Table 2 ijms-25-01173-t002:** Characteristics of all participants with confirmed amyloid status from CSF/PET analysis, and of participants without dementia with confirmed amyloid status. SD: standard deviation.

	All Participants		Non-Demented Participants	
	CSF/PET Aβ+	CSF/PET Aβ−	CSF/PET Aβ+	CSF/PET Aβ−
n	74	77	53	73
Age: mean (SD)	70.8 (8.11)	67.8 (8.67)	70.9 (7.32)	67.5 (8.63)
MMSE: mean (SD)	24.6 (4.57)	27.8 (2.15)	27.06 (1.62)	28.15 (1.54)
*APOE* *: n ε4−/n ε4+ (% ε4−/% ε4+)	20/47 (30%/70%)	49/23 (68%/32%)	15/36 (30%/70%)	48/21 (70%/30%)
Gender: male/female	33/41	35/42	24/29	32/41
Non-demented/demented	53/21	73/4	–	–
Volunteers AD patients Non-AD patients	13	53	13	53
	60	9	39	7
	1	15	1	13
Measure of amyloid:				
CSF	54	24	37	21
[18F]flutemetamol PET	17	51	15	51
[11C]PIB PET	3	2	1	1

* *APOE*: All participants: 7 *APOE* missing among Aβ+, 5 *APOE* missing among Aβ−. Non-demented participants: 2 *APOE* missing among Aβ+, 4 *APOE* missing among Aβ−. ε4− represents ε4 noncarriers and ε4+ represents ε4 carriers.

**Table 3 ijms-25-01173-t003:** Number of volunteers, AD, and non-AD patients classified as plasmatic Aβ+ or plasmatic Aβ− using the threshold of 0.0472, as determined by GMM.

	Volunteers	AD	Non-AD
Plasmatic Aβ+	115	67	10
Plasmatic Aβ−	162	3	8

**Table 4 ijms-25-01173-t004:** Number of participants classified as plasmatic Aβ+ or plasmatic Aβ− based on the plasma soluble Aβ42/Aβ40 ratio, using the 0.0472 threshold determined by GMM, and according to their amyloid status determined by CSF or PET analysis.

All Individuals	CSF/PET Aβ+	CSF/PET Aβ−
Plasmatic Aβ+	65	34
Plasmatic Aβ−	9	43

**Table 5 ijms-25-01173-t005:** Number of participants without dementia classified as plasmatic Aβ+ or plasmatic Aβ− based on the plasma soluble Aβ42/Aβ40 ratio, using the 0.0472 threshold determined by GMM, and according to their amyloid status determined by CSF or PET analysis.

Non-Demented Individuals	CSF/PET Aβ+	CSF/PET Aβ−
Plasmatic Aβ+	44	31
Plasmatic Aβ−	9	42

**Table 6 ijms-25-01173-t006:** Number of participants classified as plasmatic Aβ+ or plasmatic Aβ− based on the plasma soluble Aβ42/Aβ40 ratio, using a threshold of 0.0465, as determined by decision tree, and according to their amyloid status determined by CSF or PET analysis.

All Individuals	CSF/PET Aβ+	CSF/PET Aβ−
Plasmatic Aβ+	64	32
Plasmatic Aβ−	10	45

**Table 7 ijms-25-01173-t007:** Number of participants without dementia classified as plasmatic Aβ+ or plasmatic Aβ− based on the plasma soluble Aβ42/Aβ40 ratio, using a threshold of 0.0465 as determined by decision tree, and according to their amyloid status determined by CSF or PET analysis.

Non-Demented Individuals	CSF/PET Aβ+	CSF/PET Aβ−
Plasmatic Aβ+	43	29
Plasmatic Aβ−	10	44

## Data Availability

All data collected in the context of this study will be made available in an anonymized form upon the publication of the article.

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
