# Peer review of "Definition of a Threshold for the Plasma Aβ42/Aβ40 Ratio Measured by Single-Molecule Array to Predict the Amyloid Status of Individuals without Dementia"

_ijms, 2024, doi:10.3390/ijms25021173_

Round 1

Reviewer 1 Report

Comments and Suggestions for Authors

1.     Please, every abbreviation at the first presentation should be fully described. E.g. PET.

2.     Remove the numbers following the keywords.

3.     Why was MMSE used instead of MoCA? Especially in a developed country.

4.     Please provide a table with the participants' characteristics or revise the group description. The description of the last paragraph of the participants needs to be clarified because the present description overlaps with other groups.

5.     Was there any conflict of interest? Were all the tests bought with the funding?

6.     How were the groups distributed? How was the power of the study calculated?

7.     Were all the statistics performed in MatLab?

8.     Were healthy the participants without dementia? How did the authors avoid confounding factors in their analysis? How were assessed medications in use and supplements? How was brain volume considered before measuring the Aβ42/Aβ40 ratio?

9.     Table 1: why is the p-value being represented?

10.  Please review the discussion; there are space between the paragraphs.

Reviewer 2 Report

Comments and Suggestions for Authors

This is an interesting manuscript, with direct implication for diagnosis of AD, and thus it should be considered for publication.

However, there are a few issues that should be addressed:

1. The manuscript treats aggregation and toxicity as equivalent processes, but given the methods used (the detection is of the soluble Ab species!), these should be clearly distinguished.

Also, page 1/ line40-: on the molecular level, long Ab is not only Ab42, but rather a set of Ab-proteins, i.e., Ab40-Ab43, that could all be considered as long forms. Importantly, they all co-aggregate. Ab42 is considered more neurotoxic, but neurotoxicity stems from soluble oligomers, rather than plaques. Thus, some modification might be warranted.

2. “Single-molecule array” is a bit misleading, given that the ratio of two peptides is taken as a marker. Since the original publication (ref 11) calls it single-molecule enzyme-linked assay, aka digital ELISA, some modification of the related paragraphs (e.g., page 2) would be warranted, so that readers from various fields could have a better understanding of the process.

In addition, it would be advantageous to provide some additional information about the working principles of the specific assays that are used by the kits. Also, the reasons for why these specific kits were selected (given that there are several manufacturers), would make it more appealing to a broad readership of IJMS. In other words, if other kits to be used, how would these results be correlated, not correlated, etc, is not clear quite at this time.
